# Artificial Intelligence and Lung Cancer: Impact on Improving Patient Outcomes

**DOI:** 10.3390/cancers15215236

**Published:** 2023-10-31

**Authors:** Zainab Gandhi, Priyatham Gurram, Birendra Amgai, Sai Prasanna Lekkala, Alifya Lokhandwala, Suvidha Manne, Adil Mohammed, Hiren Koshiya, Nakeya Dewaswala, Rupak Desai, Huzaifa Bhopalwala, Shyam Ganti, Salim Surani

**Affiliations:** 1Department of Internal Medicine, Geisinger Wyoming Valley Medical Center, Wilkes Barre, PA 18711, USA; 2Department of Medicine, Mamata Medical College, Khammam 507002, India; priyathamgurram@gmail.com (P.G.); lekkala.saiprasanna@gmail.com (S.P.L.); suvidha.manne97@gmail.com (S.M.); 3Department of Internal Medicine, Geisinger Community Medical Center, Scranton, PA 18510, USA; bamgai08@gmail.com; 4Department of Medicine, Jawaharlal Nehru Medical College, Wardha 442001, India; alifya.altaf.306@gmail.com; 5Department of Internal Medicine, Central Michigan University College of Medicine, Saginaw, MI 48602, USA; s.aadil19@gmail.com; 6Department of Internal Medicine, Prime West Consortium, Inglewood, CA 92395, USA; koshiyahiren5@gmail.com; 7Department of Cardiology, University of Kentucky, Lexington, KY 40536, USA; nakeya52@gmail.com; 8Independent Researcher, Atlanta, GA 30079, USA; drrupakdesai@gmail.com; 9Department of Internal Medicine, Appalachian Regional Hospital, Hazard, KY 41701, USA; bhopalwalahuzaifa@gmail.com (H.B.); shyam.ganti@gmail.com (S.G.); 10Departmet of Pulmonary, Critical Care Medicine, Texas A&M University, College Station, TX 77845, USA; salim.surani@va.gov

**Keywords:** lung cancer, artificial intelligence, machine learning, deep learning, radiomics, screening, diagnosis, treatment, treatment response

## Abstract

**Simple Summary:**

In this comprehensive review, we aimed to summarize the advances made by artificial intelligence in the field of lung cancer screening, diagnosis, and management. We now understand the utility of AI as a tool that can supplement physicians to improve the quality of care provided, which is the core message of this review, along with the relevant literature supporting the advances.

**Abstract:**

Lung cancer remains one of the leading causes of cancer-related deaths worldwide, emphasizing the need for improved diagnostic and treatment approaches. In recent years, the emergence of artificial intelligence (AI) has sparked considerable interest in its potential role in lung cancer. This review aims to provide an overview of the current state of AI applications in lung cancer screening, diagnosis, and treatment. AI algorithms like machine learning, deep learning, and radiomics have shown remarkable capabilities in the detection and characterization of lung nodules, thereby aiding in accurate lung cancer screening and diagnosis. These systems can analyze various imaging modalities, such as low-dose CT scans, PET-CT imaging, and even chest radiographs, accurately identifying suspicious nodules and facilitating timely intervention. AI models have exhibited promise in utilizing biomarkers and tumor markers as supplementary screening tools, effectively enhancing the specificity and accuracy of early detection. These models can accurately distinguish between benign and malignant lung nodules, assisting radiologists in making more accurate and informed diagnostic decisions. Additionally, AI algorithms hold the potential to integrate multiple imaging modalities and clinical data, providing a more comprehensive diagnostic assessment. By utilizing high-quality data, including patient demographics, clinical history, and genetic profiles, AI models can predict treatment responses and guide the selection of optimal therapies. Notably, these models have shown considerable success in predicting the likelihood of response and recurrence following targeted therapies and optimizing radiation therapy for lung cancer patients. Implementing these AI tools in clinical practice can aid in the early diagnosis and timely management of lung cancer and potentially improve outcomes, including the mortality and morbidity of the patients.

## 1. Introduction

Lung cancer is the second most common cancer in both males and females, with the highest mortality worldwide, causing 21% of total cancer-related deaths [1]. Despite current screening methods, diagnosing lung cancer at an earlier stage remains challenging, accounting for a high mortality rate compared to all other cancer forms. Among the cases diagnosed, only about 20% are diagnosed with stage I, and these statistics have not changed for years [2]. Apart from the diagnosis, drawbacks in assessing the prognosis and treatment options are challenges clinicians face [3]. Although there have been significant advancements in immunotherapy and targeted treatment to treat lung cancer patients, the efficacy remains erratic, and the response rate to these is highly variable [4]. Therefore, it is necessary to look for earlier diagnostic tools for lung cancer, which are highly sensitive and specific [5]. In 2017, a couple of studies [6,7] showed the importance of utilizing the morphological features of pathological slides in diagnosing and prognosing lung cancer. This was, therefore, able to highlight the importance of computer-aided image analysis in lung cancer prognosis.

The notion of artificial intelligence (AI) was initially proposed by John McCarthy in 1956. It involves using computer systems and technology to replicate human-like intelligent behavior and critical thinking abilities [8]. In the realm of medicine, AI is divided into two main categories: virtual and physical. The virtual branch is further categorized into machine learning (ML) and deep learning (DL) [9]. ML is the remarkable capability of a system to learn from data autonomously without the need for explicit programming [10]. ML encompasses four primary categories of tasks: supervised, unsupervised, reinforcement, and active learning [11]. Supervised learning entails utilizing input data with target labels to identify a pattern. There are a variety of models for supervised ML, including Bayesian inferences, decision trees, linear discriminants, support vector machines, logistic regression, and artificial neural networks [12]. On the other hand, unsupervised learning involves identifying patterns within the input data that have not been previously labeled. Lastly, reinforcement learning involves training intelligent agents to improve their performance [13]. DL is a subset of machine learning that utilizes multiple layers simultaneously to achieve both feature selection and model fitting [14].

Over the past few years, there has been significant progress in the field of oncology, particularly in the management of lung cancer. This progress includes the development of better imaging and staging techniques, as well as the use of molecular markers to guide patient-centered treatment [15,16]. AI allows for the analysis and interpretation of intricate medical information, ultimately assisting in the diagnosis, treatment management, and prediction of outcomes for various clinical cases [17]. In clinical oncology, including lung cancer, there are currently multiple FDA approvals for AI applications [18]. Lung cancer’s heterogeneity makes it a prime field for AI applications [19]. Numerous AI-based tools have been created that are particularly useful for lung segmentation, nodule detection, and characterization, which we aim to summarize in this article.

## 2. Methods

In our review, we examined all publications from 2012 to 2023 that were published in the PubMed database. We explored the database using the following Medical Subject Heading (MeSH) terms: artificial intelligence, machine learning, radiomics, deep learning, lung cancer, lung cancer screening, lung nodule detection, lung nodule characterization, lung nodule segmentation, lung cancer diagnosis, lung cancer staging, lung cancer treatment, and treatment response. A total of 270 articles were found using the above keywords; we excluded certain studies during the screening process based on the presence of duplicate records, as well as studies with titles or abstracts that were irrelevant to our research objective. In cases where full-text papers were not available, those studies were also omitted from our review. We included a total of 69 papers in the composition of this narrative review with relevant prospective, retrospective, and review papers, with the main emphasis on the use and implementation of artificial intelligence in lung cancer. Five investigators, each with distinct backgrounds and expertise, conducted a comprehensive literature review, offering independent perspectives and resulting in a wide range of insights and interpretations. The further framework of the review is described in the figure below (Figure 1 and Figure 2) (Appendix A).

## 3. AI in Lung Cancer Screening

The National Lung Screening Trial (NLST) has revealed that early diagnosis among the high-risk population has been shown to reduce the lung cancer death rate by 20% [20]. These compelling statistics underscore the urgent need for the development of highly accurate screening methods and extensive patient education to promote early diagnosis and ultimately improve the prognosis for those affected by this devastating disease. As of March 2021, according to the U.S. Preventive Services Task Force (USPSTF), individuals aged 50 to 80 who have a history of smoking at least 20 packs of cigarettes and either smoke currently or have quit within the past 15 years are advised to undergo yearly low-dose computed tomography (LDCT) screenings for the detection of lung cancer. It is advised to stop the screening process if a person has not smoked for 15 years or has a health issue that significantly shortens their life expectancy or prevents them from having curative lung surgery [21,22].

AI models have become an integral part of the lung cancer screening process and can offer a range of benefits. These include minimizing the radiation exposure, accurately detecting and categorizing lung nodules, personalizing screening schedules, and providing LDCT interpretation in regions with a shortage of skilled radiologists [23]. Convolutional neural networks (CNNs), a class of deep-learning artificial neural networks, has shown promising results in predicting lung cancer risk by using visual imagery and clinical information retrieved from electronic medical records (EMRs). 

Numerous experimental studies have been undertaken to identify high-risk populations that can be explored further. Based on non-imaging data, CNNs have successfully identified high-risk patients and predicted a 1-year lung cancer rate with excellent accuracy, demonstrated by an overall AUC—0.90 [24]. Among the models that used imaging data, CXR-LC identified high-risk patients by relying solely on CXR findings and limited clinical data with an accuracy (AUC–0.755) comparable to the previous models like PLCO(AUC–0.751) [25], whereas Sybil (a validated model) predicted a 6-year lung cancer risk using a single LDCT scan data [26]. Another deep-learning CNN, LUMAS, predicted the 1-year lung cancer risk using previous and recent CT scans with AUC—0.94 and successfully outperformed radiologists [27]. These findings exhibit potential for large-scale screening based on EMR data, and a path toward efficient screening strategies. 

We will discuss the modalities of lung cancer screening with AI techniques under two subheadings: imaging (nodule detection, segmentation, and characterization) and non-imaging techniques. This overview aims to highlight AI’s significant contributions to lung cancer screening, offering insights into the broad spectrum of AI applications within the healthcare domain.

## 4. Imaging Techniques

LDCT is the gold standard for lung cancer screening, being the only modality shown to have mortality benefits for lung cancer patients; however, advanced predictive models are being developed, which combine CT images with innovative technologies such as AI algorithms with improved accuracy. By harnessing the power of these imaging techniques and integrating them into predictive models, researchers and clinicians were able to enhance the screening process. These efforts have improved accuracy [28] and have enabled early intervention for better patient outcomes. Several FDA-cleared AI tools have since become available for nodule detection, characterization, and reporting [29]. Although an overview of all the available tools is beyond this review’s scope, we aim to provide the status of AI in nodule detection and nodule segmentation. 

### 4.1. Nodule Detection

According to the Fleischner Society: A nodule on CT is a well or poorly defined rounded or irregular opacity and can measure up to 3 cm in diameter. These nodules can serve as early indicators of lung cancer and can be detected and characterized using CT scans [30]. Interestingly, AI has comparable results to those of experienced radiologists; their collective performance demonstrated greater accuracy in detecting pulmonary nodules on CXR and LDCT scans. Although computer-aided diagnosis (CAD) and radiomics have been traditionally used for lung cancer detection, a high false positive rate have been a significant challenge. A novel CAD system using multi-view convolutional networks (ConvNets) has shown a high detection sensitivity rate of 84.1% and 90.1% with one and four false positive tests per scan, respectively [31]. Another randomized controlled trial involving 10,476 participants showed that the AI group, which used CAD, had higher detection rates (0.52%) of actionable nodules (Lung-RADS, category 4) compared to the non-AI group (0.25%), along with improved detection of malignant nodules (0.15% vs. 0.0%) [32]. Nonetheless, DL techniques have proven to be superior at nodule detection and the prediction of lung cancer risk [29]. Notably, these techniques have even been shown to enhance the predictive value of digital chest tomosynthesis (DTS) in lung cancer detection in nodules ranging from 5 and 8 mm in size (sensitivity—0.90, PPV—0.95) [33,34]. The AI-RAD companion, a CNN prototype, could automatically detect pulmonary nodules on LDCT with improved accuracy due to its high sensitivity (=1) and specificity (=0.708). Furthermore, these outcomes contributed to an enhanced ability to predict lung cancer (AUC—0.942) [35]. Another model, the DL-CADe system, was recognized for its higher nodule detection rate (86.2% vs. 79.2%) per CT examination than that of a double reading by two radiologists [33]. Additionally, the detection of pulmonary nodules using AI was highest among standard-dose CT (AUC—0.989) compared to low-dose (AUC—0.983) and very low-dose CT scans (AUC—0.970) [36]. A DL-based automatic detection algorithm (DLAD) that detects malignant pulmonary nodules using CXR data has been developed with radiograph classification and nodule detection performances of 0.92–0.99 (AUROC) and 0.831–0.924 (JAFROC FOM), respectively [37], and findings have established the potential of neural networks as a screening modality to complement conventional methods among high-risk populations.

### 4.2. Nodule Segmentation and Characterization

Lung nodule characterization involves analyzing a nodule’s size, volume, and density to determine whether it is benign or malignant. Although an LDCT scan is the most common modality used for this purpose, AI algorithms have been found to measure these variables accurately and to track the growth of lung nodules on follow-ups. Multi-scale CNN models relying solely on raw nodule patches, without a proper definition of the morphology, were developed to capture the nodule heterogeneity, and it has achieved an 88.84% accuracy for nodule classification against noisy backgrounds [38]. AI-based radiomics models, such as SVM-LASSO, outperformed Lung-RADS in the detection of malignant nodules, utilizing two features: the bounding box anterior–posterior dimension (BB-AP) and the standard deviation of the inverse difference moment (SD-IDM). These models demonstrated an impressive accuracy of 84.6% (AUC—0.89). In contrast, Lung-RADS achieved an accuracy of 72.2% (AUC—0.77) [39].

Nodule segmentation is an image analysis technique that distinguishes and outlines lung nodules’ boundaries from the surrounding thoracic tissue. This allows us to accurately measure the volume of the nodule, which is vital in determining its size. It also enables the measurement of the nodule density or attenuation, aiding in the determination of its composition and potential malignancy. This significantly enhances the test’s ability to distinguish benign and malignant nodules, which further aids in making subsequent treatment decisions. The manual segmentation of the nodules is tedious, and its semi or full automation through training AI models has significantly improved the efficiency and accuracy in characterizing the nodules detected. SD-Unet, a deep-learning model used for biomedical segmentation, trained to classify the image voxels, resulted in improved segmentation [40]. In addition to CNN models, Soliman et al. [40] developed a spatially non-uniform joint 3D Markov–Gibbs random field (MGRF). This method effectively segmented nodules by integrating two visual appearance submodels with an adjustable lung shape submodel. It exhibited accurate results with a DICE similarity coefficient of 98.4 ± 1.0% and 99.0 ± 0.5% upon validation with an external database [41].

#### Non-Imaging Techniques

The analysis of body fluids, generally blood, is a non-invasive method widely used to detect biomarkers and tumor markers. This approach plays a crucial role in the comprehensive management of cancer patients, encompassing screening, diagnosis, treatment, and patient follow-up. Emerging as potential biomarkers for lung cancer screening are autoantibodies, complement fragments, miRNA, tumor DNA, and serum proteins [42]. While these biomarkers show promise, their sensitivity and specificity remain limited, classifying them as supplementary screening tools. This approach plays a vital role in identifying, managing, and keeping track of patients with cancer. Autoantibodies, complement fragments, miRNA, tumor DNA, and serum proteins have emerged as potential biomarkers for lung cancer screening [42]. Biomarkers are considered supplemental screening tools due to their limited sensitivity/specificity. The utilization of predictive models that incorporate multi-biomarker panels, CT-scan images, and AI has proven to be a game-changer, greatly enhancing both the specificity and accuracy in the early detection and diagnosis of lung cancer.

In the test phase, the implementation of Artificial Neural Networks (ANNs) alongside serum protein panels (β2-microglobulin, CEA, gastrin, CA125, NSE, sIL-6R, and three metal ions: Cu^2+^/Zn^2+^, Ca^2+^, and Mg^2+^) demonstrated a commendable prediction rate of 85%. Additionally, incorporating clinical parameters (such as symptoms, risk factors, smoking, and kitchen environment) resulted in an increased prediction rate of 87.3% [43]. On combining the Pulmonary Nodules Artificial Intelligence Diagnostic System (PNAIDS), which analyzes CT images, along with tumor markers (TM), the predictive models had the highest specificity (96.1%), whereas integration with circulating abnormal cells has shown to have a specificity of 94.1% [44]. These compelling findings signify the potential of these approaches as novel screening tools for the early detection of lung cancer.

In conclusion, incorporating AI into lung cancer screening has enormous potential to revolutionize early detection and enhance patient outcomes. However, further research and integration of AI systems into clinical practice are required to ensure their safety, reliability, and widespread adoption. This is possible by the ongoing advancements and collaborations between the medical and AI communities.

## 5. AI in Lung Cancer Diagnosis

Lung cancer diagnosis primarily relies on a CT scan and tissue biopsy, which can lead to misdiagnosis and omissions [45]. Enhancing the sensitivity and specificity of non-invasive biomarkers is crucial. Factors like tumor location, pathology type, metastasis presence, and complications make diagnosis challenging [46]. AI models have become an effective tool in lung cancer diagnosis, improving the accuracy, stability, and efficiency [45,47]. This review covers the applications of AI models in diagnostic imaging, pathology tests, and biomarkers (Table 1).

### 5.1. Diagnostic Imaging

It involves using CT and PET-CT (positron emission tomography-computed tomography) of the chest to find an abnormal mass or tumor in the lung [45]. Screening using CT scans takes time and is subject to variation between people. With the growing popularity of AI, the medical field has recognized its impact in assisting the use of diagnostic imaging [45]. Ardila et al. [26] developed a deep learning algorithm for detecting lung cancer by low-dose CT scan and achieved a striking AUC of 94.4%. Another study examined the CT scans of 200 lung nodules with an AUC of 0.72 [48]. Additionally, a study investigated the use of ML for detecting lung cancer by FDG-PET imaging and achieved a sensitivity of 95.9% and 91.5% and a specificity of 98.1% and 94.2% with standard dose and ultralow dose, respectively. These findings indicate that ML modules may help detect lung cancer even at a very low radiation exposure of 0.11 mSv [50]. In a meta-analysis by Liu, the combined sensitivity, specificity, and sum of area under the combined subject operating characteristic (SROC) curve of the AI-aided diagnosis system for lung cancer diagnosis by using CT images were 87%, 87%, and 93%, respectively [44]. In another meta-analysis, which included nine NSCLC studies, the pooled sensitivity and specificity were 78% and 71%, respectively, and the AUROC of radiomics was 0.78 (95% CI 0.73–0.83) [50].

A study by Sun [50] included 395 pure ground glass nodules (pGGNs) from 385 patients who were randomly assigned to a training set (*n* = 277) and a validation set (*n* = 118). Based on the radiomics, a nomogram was developed on the RAD score, margin, speculation, and nodule size. The combined radiographic–radiomics model (AUC 0.77; 95% CI, 0.69–0.86) predicted the invasiveness better than the radiographic model (AUC 0.71; 95% CI, 0.62–0.81) in the validation set. This model may be used to evaluate invasive prediction in patients with pGGNs [51]. To validate the efficiency of radiomics, another Chinese retrospective study evaluated 100 patients with solitary sub-solid nodules confirmed pathologically with either minimally invasive (MIA) or invasive adenocarcinoma (IAC). They constructed an integrated model using CT-based findings like nodule size, shape, margins, and radiomic signatures. This model showed good differentiation in the training set (AUC 0.943) and validation set (AUC 0.912) [53]. From these findings, we conclude that machine-learning features can be integrated with CT-based subjective findings to improve the accuracy of tumor differentiation and their invasiveness.

Another study, which included 301 lung carcinoma images from CT scans, correctly detected lung cancer using convolutional deep neural network methods with 0.93 sensitivity, 0.82 precision, and a 0.87 F1 score. This CNN model further differentiated small cell lung carcinoma, adenocarcinoma, and squamous cell lung carcinoma, with sensitivity, specificity, and F1 scores of 0.90, 0.44, and 0.59, respectively [54]. Lastly, Saad et al. achieved an AUC of 0.93 in differentiating NSCLC and peripherally located small cell lung cancer (SCLC) by using radiomics [58]. Physicians rely on pathological analysis to reveal these phenotypic variations, which require invasive methods such as biopsy and resection samples [7]. But AI-mediated imaging can help detect subtypes, which is non-invasive and can assist in starting early treatment.

### 5.2. Histopathological Diagnosis

This often refers to histological examination through bronchoscopy or percutaneous puncture biopsy, the gold standard for lung cancer diagnosis. Manual reading is difficult when assessing the pathological type of lung cancer because of the many subtypes. In a study by Yu, they used 2480 histopathological images from squamous cell carcinoma and adenocarcinoma of the lung and successfully differentiated malignant tumors from healthy tissues with an AUC of 0.81 [7]. Teramoto et al. examined 298 images using deep CNNs and classified adenocarcinoma, squamous cell carcinoma, and small cell lung cancer with an accuracy of 89%, 60%, and 70%, respectively, which was higher than the accuracy of cytotechnologists and pathologists [55]. In another prospective study, the prediction model that included clinical information (age and smoking history), radiological features of lung nodules (nodule diameter, nodule count, upper lobe location, malignant sign at the nodule edge, and sub-solid status), and LDCT data from AI analysis and liquid biopsy gave the best detection results in the training group (a sensitivity of 89.53%, specificity of 81.31%, the area under the curve [AUC] = 0.880). This can be applied to improve early lung cancer diagnosis while sparing patients with benign features from harmful surgery [59]. AI-mediated histopathological diagnosis will increase pathologists’ productivity and will significantly decrease misdiagnosis [46].

### 5.3. Biomarkers

The most common biomarkers predicting lung cancer are Rb, K-RAS, EGFR, c-MET, TP53, ALK, and PDL1 [19,56]. Though several potential biomarkers have been identified, their clinical utility remains limited because of a lack of consistency in diagnosis and predicting prognosis. Now, AI-mediated proteomics is trying multiple biomarker panels for the better detection of different types of lung cancer. Coudray et al. [56] anticipated that specific gene mutations would modify the framework of lung cancer cells in the section images; they predicted the ten most common mutant genes in adenocarcinoma by training neural networks. Pathological images predicted six of them (KRAS, STK11, TP53, EGFR, SETBP1, and FAT1) with an accuracy of 73.3–85.6% [60]. In another study, Zhong et al. [60] measured the five most predictive antibody markers, tentatively, paxillin, SEC15L2, BAC clone RP11-499F19, XRCC5, and MALAT1 in 23 stage 1 NSCLC patients and 23 risk-matched control samples. All 46 samples were used as a training set and were combined in a logistic regression model, yielding an AUC of 0.99, a 91.3% sensitivity, and a 91.3% specificity [57]. A study tested a biomarker panel composed of Cyfra 21.1, CEA, CA125, and CRP in 63 patients with lung cancer and 87 noncancer patients. This panel correctly classified 135/150 subjects. In the training, validation, and testing phases, the accurate classification rate of the lung cancer patients was 88.9%, 93.3%, and 90%, respectively [60]. Furthermore, research has shown that using a diagnostic model on lung cancer that includes human epidermis secreting protein 4 (HE4), secreting vascular cell adhesion molecule-1 (sVCAM-1), sarcosine (TTR), apolipoprotein A2 (ApoA2), sarcosine (TTR), in conjunction with the carcinogenic antigen CEA can greatly improve lung cancer detection accuracy. With a sensitivity of 93.33% and a specificity of 92.00%, this model achieved an AUC value of 0.988, suggesting a strong prediction accuracy [46]. As there is no universal biomarker panel for lung cancer, optimized panels must be tested and validated in each population before being applied in the clinical setting [61]. These results imply that deep learning algorithms may help pathologists detect cancer subtypes and genetic mutations [60]. As a result, based on available data from various studies, AI recognition technology may aid clinicians in screening and diagnosing early lung cancer [45].

## 6. AI in Lung Cancer Staging

Accurate staging of lung cancer can aid in creating the most appropriate treatment strategies and prognosis. Non-small cell carcinoma can be staged from I through IV using the clinical, radiological, and biopsy findings that are available, and small cell carcinoma is divided into limited and extensive diseases [45,62,63].

Lung cancer is typically staged using the TNM (tumor, node, metastasis) classification. The prompt staging of lung cancer requires imaging techniques including CT and PET. Most lung cancers are typically detected at an advanced stage and may have a dismal prognosis [64]. AI is adept at handling a sizable amount of computational and repetitive labor work, making it suited for supporting medical professionals in assessing diseases with a high visual component [65].

AI might accelerate the accurate staging of lung cancer and curb the time-consuming tasks of reading pathology slides and CT scans. Using AI as a second reader for PET and CT reading lessens the work required of radiologists and improves the nodule detection precision.

Lung cancer staging is dependent on the results of PET and CT scans. PET scanning facilitates studying the extent of spread of metastatic cancer. A CT scan assists in determining the extent of local extension [66,67]. The accuracy of the results depends heavily on the radiologist’s competence, and the CT pictures of lung nodules are complex. Manual film interpretation often results in inaccurate or missing diagnoses, rendering early lung cancer diagnosis more challenging. As a result, restricting the observation error is a crucial tactic [68,69]. AI could help to solve these observational errors.

Using AI for image analysis may improve the tumor staging precision. CNNs can predict the anatomical locations of metastatic lesions using the multiplanar reconstruction of PET and CT scans. The study demonstrated how the application of AI in image processing could lead to more precise, early stage diagnoses and increase physicians’ effectiveness [19,66].

Based on the full CNN model, a researcher proposed a new model named DFCnet, which is overall more accurate than the CNN model. Both CNN and DFCNet had overall accuracy rates of 77.6% and 84.58%, correspondingly. The precision of the proposed model for lung cancer stage detection and classification is demonstrated by experimental data. These findings show the method’s potential for helping the physician precisely and efficiently improve the cancer stage classification [69,70].

Implementing new AI base models and computational systems will certainly boost the physician efficiency while managing lung cancer, and it will also assist in classifying lung cancer staging and enhancing prognosis predictions.

## 7. AI in Lung Cancer Treatment

Recently, AI has been recognized as a potent ally in lung cancer treatment. AI models like DL and radiomics have shown remarkable potential in assisting clinical decision-making processes. They offer a quantitative interpretation of patients’ information and could effectively navigate the dynamic nature, individual differences, and inherent heterogeneity associated with lung cancer [71] (Table 2).

### AI in Treatment Recommendations

Lung cancer lesions are treated with surgical and non-surgical modalities like radiation, chemotherapy, and immunotherapy. Radiotherapy plays a crucial role in lung cancer treatment, and AI has shown promise in optimizing this modality. ML systems can utilize high-quality data generated by radiotherapy systems, including CT scans and treatment histories, to enhance radiation treatment planning. ML algorithms can optimize radiation beam angles, predict dose–volume histograms, monitor radiation levels and toxicity, and develop clinical decision support tools [73]. These integrative predictive models offer the potential for personalized radiation treatments with improved safety and efficiency.

Luo et al. [6] suggested an integrated learning collaborative filtering technique in ML to simplify the selection of therapeutic medications for individualized lung cancer therapy and to identify possible drug candidates. Another approach, QUANIC, utilizes large-scale multimodal and longitudinal data to develop personalized models for the immunotherapy response and resistance in lung cancer [46].

Immunotherapy presents challenges in patient selection and predicting treatment response. However, ML techniques, along with radiomics, have shown promise in improving patient selection and predicting treatment outcomes by providing non-invasive insights into the tumor and its microenvironment. ML algorithms have been employed to identify the determinants of tumor immunogenicity and develop scoring schemes for predicting the response to immune checkpoint inhibitors (CPIs). Additionally, predictive radiomic features extracted from CT images have been used to predict treatment responses [73]. Kureshi et al. developed a data-driven model for predicting the tumor response to EGFR-TKI therapy in advanced NSCLC patients. The model achieved a predictive accuracy of 76% by considering various factors, including clinical history, environmental risk factors, and the EGFR mutation status. Additionally, Liu et al. developed a predictive model for overall treatment recommendation with 65.8% consistency [73,75].

Radiomics, which involves a quantitative analysis of imaging features, has emerged as a promising tool for predicting the individual responses to immunotherapy. By capturing the tumor heterogeneity and immune infiltration from CT images, radiomics enables the development of radiomic signatures that are associated with the treatment response. ML markers based on radiomics have successfully predicted immunotherapy responses in patients with NSCLC. Integrating radiomics with DL further enhances its potential for selecting responsive patients for immunotherapy [71].

AI has demonstrated its potential for enhancing the surgical risk prediction and assisting in treatment decisions. IBM Watson for Oncology (WFO) is an AI system that aids doctors in extracting crucial information from medical records, presenting relevant evidence, and exploring treatment options. WFO has shown a high level of consistency with the recommendations provided by the oncology committees, highlighting its value in supporting multidisciplinary teams [46].

WFO was used in China and Korea to provide accurate treatment recommendations for lung cancer patients. It showed high potential and concordance with the multidisciplinary team (MDT) recommendations, particularly in the advanced stages of cancer [18,76]. However, there are distinct challenges in applying WFO between the USA and other countries due to the variations in genetic mutation rates, treatment protocols, drug availability, and coexisting diseases. Adapting WFO to regional-specific factors and individualized patient information would optimize its applicability and improve the treatment consistency [76].

In addition to treatment recommendations, AI also has the potential to support drug development by repurposing existing drugs for new uses and identifying potential drug candidates for further investigation. In one study, a DL algorithm using transcriptomic and chemical structures identified pimozide as a candidate for treating NSCLC, which was validated in vitro. Another study used neural networks to predict the postoperative outcomes in NSCLC patients, achieving high accuracy in predicting the cardio-respiratory toxicity and postoperative complications. These findings demonstrate the value of AI in drug development and patient risk assessment [78].

## 8. AI in Predicting Treatment Outcomes

AI has been shown to have the capacity to play a role in medical decisions by predicting treatment responses, including survival and adverse events, and helping to choose a group of patients to receive a specific treatment [19]. Dercle et al. reported that an AI model based on the CT-based radiomic characteristics and random forest algorithm accurately predicted the treatment response of various therapies like nivolumab, docetaxel, and gefitinib [81].

DL models have shown promising ability in identifying therapy response and prognosis. Specifically, these models have successfully predicted the EGFR mutation probability and patient response to EGFR-tyrosine kinase inhibitors (TKIs) and CPIs. By accurately identifying patients at different risks of progression, these AI models could aid in treatment decision-making and improve patient outcomes [80,82,83].

Additionally, by identifying specific radiomic features associated with local failure, tumor recurrence, and chemotherapy response, radiomics-based models can guide treatment decisions and predict treatment outcomes [84]. Furthermore, integrating multi-omics data through AI in precision medicine holds great promise. Radiomics-based AI models have demonstrated the ability to predict PD-L1 expression levels by combining radiomic images with clinical data. These models have also shown prognostic performance in predicting progression-free survival and clinical benefit in immunotherapy candidates [85].

ML applications have also been used to predict early death following curative intent chemoradiation and failure in early stage NSCLC patients treated with stereotactic body radiation therapy, and this can be used to educate patients about possible treatments and optimize care [73]. AI has also shown potential for incorporating serial imaging data to track tumor changes over time. By leveraging DL methods and recurrent neural networks (RNN), AI can analyze longitudinal data from post-treatment CT scans and provide valuable insights into phenotypic characteristics and treatment response [86].

## 9. Limitations and Future Perspectives

One of the limitations of AI is the lack of large datasets of clinical data to train the model, which can hinder its performance. While public datasets like LIDC-IDRI and LUNA 16 are available, they have certain challenges, like data variability as they contain images from limited centers and may not represent the entire population, a lack of clinical information, and potential data annotation errors. AI-driven radiomics and DL could become universal through collaboration among multiple healthcare institutions to create an interpersonal, standardized dataset that includes diverse patient populations, various stages of lung cancer, and longitudinal information. This can be a valuable resource for training AI models and improving their generalizability. The standardization of real-world data (RWD) is also an exciting step in the development and training of AI models since the FDA passed the 21st Century Cures Act which has a comprehensive framework to use postapproval data for newly approved drugs; however, with RWD being easily accessible and the lack of data quality due to heterogeneity leading to difficulty in data interpretation, the lack of reproducibility and replicability should be cautioned and considered before large-scale implementation [51,87]. By reducing the inconsistencies and errors, this standardization process will improve the reliability of AI and RWD applications and prove as an asset for the large-scale use and better training of the AI model. There are challenges to implementing AI in clinical practice, like a lack of resources and proper training and education among healthcare professionals. Healthcare systems should implement substantial infrastructure and training for all healthcare personnel to effectively use and interpret AI tools. In rapidly evolving healthcare systems and workflows, these AI tools should be updated regularly for seamless performance. Programming interfaces should be developed that allow AI algorithms to integrate with EHR systems. This would enable the real-time data exchange between AI tools and clinical systems, facilitating efficient decision-making for lung cancer management. Regular feedback should be gathered from oncologists, radiologists, and other specialists to refine the AI model’s performance. Meanwhile, ensuring patient privacy, data security, data ownership, and compliance with regulations such as HIPAA can be complex. A framework with clear guidelines for the acceptance and deployment of AI models in healthcare should be established to ensure patient safety and ethical standards in handling the data. Some of the predictions of AI in lung cancer treatment can be challenging for clinicians to interpret and extract meaningful insights from because of the lack of transparency in explaining the rationale behind those decisions, which needs to be addressed. Lastly, incorporating AI models to use clinical data in context with the imaging findings to further guide the physician in clinical outcomes of the patient, aiding in shared decision-making, remains a future prospect of AI in lung cancer management.

## 10. Conclusions

The emergence of AI has brought about a significant transformation in the management of lung cancer. Its wide-ranging applications in screening, early diagnosis, treatment selection, and prediction of treatment response offer promising possibilities for enhancing patient outcomes and driving the progress of precision medicine. Despite AI’s remarkable potential for lung cancer, it is crucial to acknowledge the challenges and limitations accompanying its implementation. Addressing the data quality, interpretability of models, and ethical considerations becomes imperative to ensure the successful integration of AI into clinical practice. By navigating these obstacles, we can unlock the full potential of AI and pave the way for a more effective and personalized approach to lung cancer care.

## Figures and Tables

**Figure 1 cancers-15-05236-f001:**
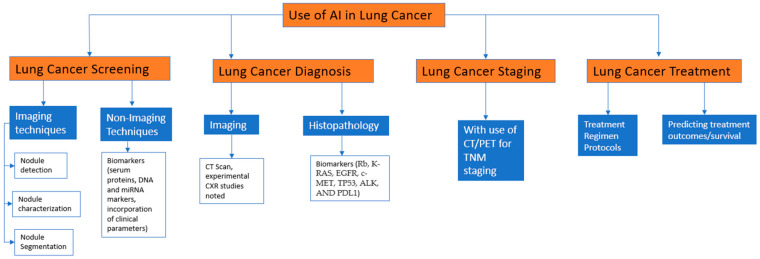
Implementation of AI in the field of Lung cancer and our layout of presentation in the review.

**Figure 2 cancers-15-05236-f002:**
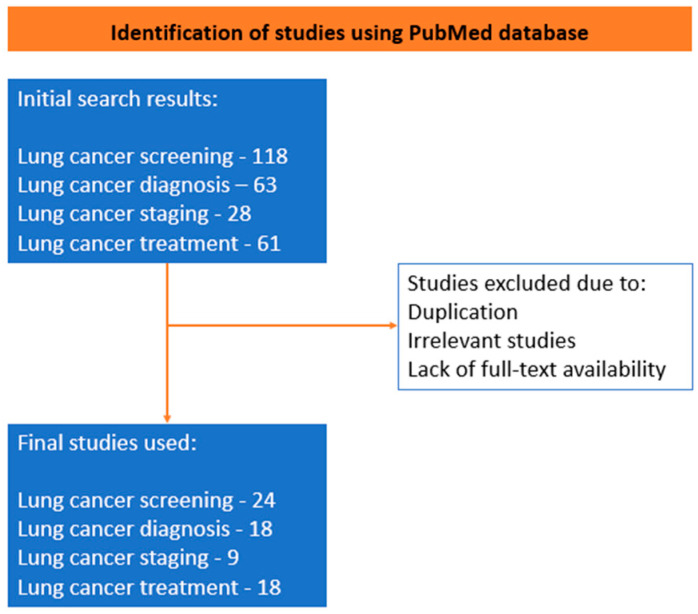
Algorithm for search stratergy for the comprehensive review.

**Table 1 cancers-15-05236-t001:** Studies showing the Diagnosis of lung cancer using AI models.

Author, Year	Dataset	AI Algorithm	Outcomes	Results
Ardila et al., 2019 [26]	Low-dose CT scan	Deep learning algorithm	Diagnosis of lung cancer	AUC = 0.94
Delzell et al., 2019 [47]	CT scan of 200 lung nodules	Radiomics	Verify nodules as benign or malignant	AUC = 0.72
Schwyzer et al., 2018 [48]	FDG-PET imaging	Deep machine learning	Diagnosis of lung cancer using ultra-low-dose PET scans	Sensitivity = 95.9% Specificity = 98.1%
Liu et al., 2023 [44]	Images and radiological features of 5251 patients from 14 studies	ANN SVM	Diagnosis of lung cancer	Sensitivity = 87% Specificity = 87%
Zheng et al., 2022 [49]	CT images of 9 NSCLC studies	Radiomics Deep learning	To diagnose whether patient had NSCLC	AUROC = 0.78
Sun et al., 2020 [50]	Pure ground glass nodules of 385 patients	Radiomics	Invasiveness prediction	AUC = 0.77
Feng et al., 2019 [51]	Sub-solid nodules of 100 patients	Radiomics	Differentiate minimally invasive and invasive adenocarcinoma	AUC = 0.912
Avanzo et al., 2020 [52]	Nodules of low-dose CT scan	SVM	Differentiate adenocarcinoma from focal pneumonia	Accuracy = 87.6%
Aydin et al., 2021 [53]	301 lung cancer CT scans	CNN	Differentiate into squamous cell, adenocarcinoma, and small cell carcinoma	Sensitivity = 90% Specificity = 44%
Chen et al., 2020 [54]	CT radiomics of 69 lung cancer patients	Radiomics	Differentiate NSCLC from SCLC	AUC = 0.93
Yu et al., 2016 [7]	2480 histopathological images of lung adenocarcinoma	SVM Random forest	Distinguish malignant tumors from healthy tissue	AUC = 0.81
Teramoto et al., 2017 [55]	298 histopathological images	Conventional deep neural networks	Classified adenocarcinoma, squamous cell carcinoma, and small cell carcinoma	Accuracy = 89%, 60%, 70% respectively
Coudray et al., 2018 [56]	Pathological images of adenocarcinoma	Conventional deep neural networks	Predicted 10 most prevalent genes in adenocarcinoma	Accuracy = 73.3%–85.6%
Flores-Fernandez et al., 2012 [57]	Serum biomarkers of 63 lung cancer patients	Artificial neural network modeling	Correctly classifying lung cancer patients based on biomarker panel	Correct classification rate = 93.3%

SVM—support vector machine, ANN—artificial neural network, CNN—conventional neural networks, NSCLC—non small cell lung cancer, SCLC—small cell lung cancer.

**Table 2 cancers-15-05236-t002:** Studies showing implementation of AI for treatment of lung cancers.

Study	Dataset	AI Methods	Predicted Outcomes	Performance Metrics/Results
Coroller et al., 2016 [72]	NSCLC	Radiomics-based model	Predicting pathologic complete response to chemoradiation	AUC: 0.61
Kureshi et al., 2016 [73]	NSCLC	Radiomics-based model	Predicting response to EGFR-TKI therapy	AUC: 0.76
Tian et al., 2021 [74]	NSCLC	Radiomics and deep learning	Predict response to PD-1 and PD-L1 immunotherapy	AUC: 0.71
Liu et al., 2018 [75]	NSCLC and SCLC	WFO	Feasibility in treatment recommendation	Consistency 65.8%
Kim et al., 2020 [76]	NSCLC and SCLC	WFO	Treatment concordance between MDT and WFO	Concordance 92.4%
Santos-Garcia et al., 2004 [77]	NSCLC	Neural Network	Predict postoperative cardio-respiratory	AUC: 0.98
Dercle et al., 2020 [78]	NSCLC	Radiomics-based model	Treatment Sensitivity of Nivolumab	AUC: 0.77
			Treatment Sensitivity of Docetaxel	AUC: 0.67
			Treatment Sensitivity of Gefitinib	AUC: 0.82
Zhang et al., 2021 [79]	Adenocarcinoma	Radiomics-based model	Predicting EGFR mutation for targeted therapy	AUC: 0.84
Mu et al. 2020 [80]	NSCLC	Deep learning models	Predicting EGFR mutation for targeted therapy	AUC: 0.83

NSCLC: non-small cell lung carcinoma, SCLC: small-cell lung carcinoma, MDT: multidisciplinary team; WFO: Watson for Oncology; AUC: area under the curve.

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
