# Peer review of "Artificial Intelligence and Lung Cancer: Impact on Improving Patient Outcomes"

_cancers, 2023, doi:10.3390/cancers15215236_

Round 1

Reviewer 1 Report (Previous Reviewer 4)

Comments and Suggestions for Authors

We thank the authors for addressing our final concerns.

Author Response

Thank you for your comments that helped us improve our manuscript. 

Reviewer 2 Report (Previous Reviewer 3)

Comments and Suggestions for Authors

1. Please refer to the literature below to improve Figure 1.

: https://systematicreviewsjournal.biomedcentral.com/articles/10.1186/s13643-021-01626-4

2. The following papers cited in Tables 1 and 2 are not included in the references. Cited papers must be in references. Overall, a reexamination is needed.

Fengel et al., 2019, Avanzo et al., 2020, Termanto et al.2017, Coroller et al. 2016, Tian et al. 2021, Santos-Garcia et al. 2004, Zhang et al. 2021

3. In order to effectively implement artificial intelligence, the standardization of medical data such as RWD (real-world data) must be further developed. Please include these aspects in your discussion.

4. For the completeness and readability of the research, it would be a good idea to diagram the main results in figures.

Comments on the Quality of English Language

Minor editing of English language required

Author Response

  1. Please refer to the literature below to improve Figure 1.

: https://systematicreviewsjournal.biomedcentral.com/articles/10.1186/s13643-021-01626-4

Thank you for your comments, since we have conducted a comprehensive review, we didnot have a specific methods section, however, we have listed our initial search strategy and exclusion criteria in text and have also added a figure with it, please review and hope its up to the mark. 

  1. The following papers cited in Tables 1 and 2 are not included in the references. Cited papers must be in references. Overall, a reexamination is needed.

Fengel et al., 2019, Avanzo et al., 2020, Termanto et al.2017, Coroller et al. 2016, Tian et al. 2021, Santos-Garcia et al. 2004, Zhang et al. 2021

Thank you for your comments. We have added the citations at the end. 

  1. In order to effectively implement artificial intelligence, the standardization of medical data such as RWD (real-world data) must be further developed. Please include these aspects in your discussion.

Thank you for your comments, we have added that in future perspective section.

  1. For the completeness and readability of the research, it would be a good idea to diagram the main results in figures.

We have added two different synopsis pictures we have worked on, we request the reviewer to help determine which one should be used for the publication. Thank you for the suggestion, we greatly appreciate it. 

We have also made changes to language. Thank you for the comments. 

This manuscript is a resubmission of an earlier submission. The following is a list of the peer review reports and author responses from that submission.

Round 1

Reviewer 1 Report

Comments and Suggestions for Authors

The topic is not new.

The article shows a very superficial knowledge of the subject by the authors

Title

remove ITS

Intro"This review aims to present a comprehensive overview of the current applications of 121 AI in two primary domains of lung cancer screening: Imaging and Non-imaging techniques." it's unclear if the review is only on AI applications on lung cancer screening or on lung cancer in general

" radiomics have been traditionally used for lung cancer detection" radiomics is not used for cancer detection but for nodule characterization!!!

"Low dose Computed tomography (LDCT) is the gold standard for lung cancer screening;" LDCT is the only technique proven effective in mortality reduction in lung cancer screening programs!

", due to their limitations" their of whom?

" due to their limitations in terms of sensitivity and specificity," are you sure? provide an appropriate ref

"Artificial Intelligence (AI) " once you have explained the abbreviation, hìjust use it

" By harnessing the power of these imaging techniques and integrating them into predictive models, researchers and clinicians were able to enhance the screening process" The sentence is too vague: are you talking about CAD for increased accuracy or about the possibility of AI for nodule characterization?

"Pulmonary nodules are abnormal growths in the lung" This definition is not standard and needs to be modified

"a high false positive rate was a significant challenge" related to which factors?

"A novel CAD system using multi-view convolutional networks (ConvNets) has shown a high detection sensitivity rate of 84.1% and 90.1%" It has 2 different detection sensitivity rates????

"with reduced false positives" reduce compared to what?

"Nonetheless, deep-learning techniques have proven to be superior at nodule detection and the prediction of lung cancer risk. These techniques utilize rib and vessel suppression which significantly improve the detection of pulmonary nodules"

improve is improves. and, moreover, not all DL techniques remove ribs and vessels

"Notably, these techniques have even been shown to enhance the predictive value of CXR" by removing ribs and vessels? I do not think so. There is quite a confusion in AI systems for CAD purposes developed for CXR and CT

"and digital chest tomosynthesis (DTS)" The authors are talking about lung cancer screening. I haven't heard of recognized screening programs using DTS so far...

"Among these advancements is the AI-RAD companion, CNN, due to its high sensitivity" Honestly I can get the meaning: CNN is an AI-RAD companion??? Maybe the term CNN is not clear to the authors

"(=1) and low false-positive nodule rate using 151 CT scan data, which has garnered significant attention" what do you mean by significant attention??

"Additionally, the detection of 154 pulmonary nodules using AI was highest among standard-dose CT(AUC-0.989) com- 155 pared to low-dose (AUC-0.983) and very low-dose CT scans (AUC-0.970)" when talking about lung cancer screening, only LDCT is used, not standard CT. There are a lot of studies about the performance of DL-based CAD in lung nodule detection, why did the authors choose to talk about non-pertinent studies?

"DL-based detection algorithm (DLAD) " do you mean DL based CAD?

" conventional methods among high-risk populations." I cannot get if conventional methods involve machine learning or radiological assessment

"nodule patches"?

"utilizing 2CT features: the bounding box anterior-posterior dimension (BB-AP) and the standard deviation of inverse difference moment (SD-IDM)." These are not CT features

"Lung nodule segmentation plays a crucial role to clearly distinguish the areas and limits of the lung field from surrounding thoracic tissue" this sentence has no meaning

"Manual segmentation of the airway is tedious" if you are working on lung nodules, you don't have to segment airways but just the nodules

"Although CNN is considered the gold standard for nodule segmentation" this is not true, support with references

Non-Imaging Techniques:

When talking about lung K screening, non-imaging models should include not only blood tests but also the history and amount of smoke (current or not), emphysema, and family history... these factors are essential and not mentioned in the manuscript

"Factors like tumor location, pathology type, metastasis presence, and complications make diagnosis challenging" what do you mean

"AI in lung cancer detection" Are you still talking about screening or not?

"It involves using chest images," chest images are a diagnostic technique.

" MRI," MRI is not the gold standard for lung K investigation

", low dose computed tomography (LDCT)" use the abbreviation for the whole text

"The prompt staging of lung cancer requires the use of imaging techniques which 314 include computed tomography (CT) and positron emission tomography (PET) and CXR." Nobody use CXR for lung k staging

" Lung nodules are the most prevalent imaging signs of lung cancer in the early stages, which makes manual CT scan reading more challenging" repetitive and already stated previously

"AI might accelerate the accurate staging of lung cancer and curb the time-consuming tasks of reading pathology slides, CT scans, and CXR. The use of AI as a second reader for PET, CT, and CXR" again, CXR is not a technique for lung k staging

"Using Convolutional neural networks (CNN) as a model," again, you have already used in the text CNN different times

Comments on the Quality of English Language

Careful check is needed

Author Response

The topic is not new.

Thank you for your comments, we understand that with a review article novelty remains challenging, but our goal of the paper was to provide a more concise summary and updated information for the readers.

The article shows a very superficial knowledge of the subject by the authors

Thank you for your comments, we thank the reviewer for their insight.  This manuscript/review is designed for the education and awareness of the clinicians not only who are practicing in the specialiized area but also to reach and orient the latest happening in the field of Lung Cancer and AI for the clinicians who are not utilizing the AI in their armamentarium.

Title

remove ITS

Thank you for your comment, we removed it.

Intro"This review aims to present a comprehensive overview of the current applications of 121 AI in two primary domains of lung cancer screening: Imaging and Non-imaging techniques." it's unclear if the review is only on AI applications on lung cancer screening or on lung cancer in general

Thank you for the comments, we have replaced the sentence with: We will discuss the modalities of lung cancer screening with AI techniques under two subheadings.

" radiomics have been traditionally used for lung cancer detection" radiomics is not used for cancer detection but for nodule characterization!!!

Thank you for the comment, the paper we have cited uses radiomics for both detection and characterization – we have modified our statement accordingly.

"Low dose Computed tomography (LDCT) is the gold standard for lung cancer screening;" LDCT is the only technique proven effective in mortality reduction in lung cancer screening programs!

We agree with the above statement, we have made modifications and have added the appropriate reference.

", due to their limitations" their of whom?

 We have removed the above statement

" due to their limitations in terms of sensitivity and specificity," are you sure? provide an appropriate ref

Given the above comment, we have made changes to the language and removed the above statement

"Artificial Intelligence (AI) " once you have explained the abbreviation, hìjust use it

 Done, we apologize for the repetition.

" By harnessing the power of these imaging techniques and integrating them into predictive models, researchers and clinicians were able to enhance the screening process" The sentence is too vague: are you talking about CAD for increased accuracy or about the possibility of AI for nodule characterization?

"Pulmonary nodules are abnormal growths in the lung" This definition is not standard and needs to be modified

Thank you for pointing it out, we have made the changes with the definition from the Fleischner guidelines.

"a high false positive rate was a significant challenge" related to which factors?

"A novel CAD system using multi-view convolutional networks (ConvNets) has shown a high detection sensitivity rate of 84.1% and 90.1%" It has 2 different detection sensitivity rates????

Thank you for pointing it out, we have mentioned the simultaneous FP rate, so the two sensitivities make sense – they have that as they continued testing and compared the FP rate. A novel CAD system using multi-view convolutional networks (ConvNets) has shown a high detection sensitivity rate of 84.1% and 90.1% with 1 and 4 false positive tests per scan respectively.

"with reduced false positives" reduce compared to what?

 We have made changes to the sentence – the reduction was in context with the sensitivity.

"Nonetheless, deep-learning techniques have proven to be superior at nodule detection and the prediction of lung cancer risk. These techniques utilize rib and vessel suppression which significantly improve the detection of pulmonary nodules"

improve is improves. and, moreover, not all DL techniques remove ribs and vessels

Thank you for the comment we have removed the statement.

"Notably, these techniques have even been shown to enhance the predictive value of CXR" by removing ribs and vessels? I do not think so. There is quite a confusion in AI systems for CAD purposes developed or CXR and CT

We understand the confusion and have removed the citation.

"and digital chest tomosynthesis (DTS)" The authors are talking about lung cancer screening. I haven't heard of recognized screening programs using DTS so far...

We understand that these tests are not standard, but rather experimental as per our cited study: Chauvie S, De Maggi A, Baralis I, Dalmasso F, Berchialla P, Priotto R, Violino P, Mazza F, Melloni G, Grosso M; SOS Study team. Artificial intelligence and radiomics enhance the positive predictive value of digital chest tomosynthesis for lung cancer detection within SOS clinical trial. Eur Radiol. 2020 Jul;30(7):4134-4140. doi: 10.1007/s00330-020-06783-z. Epub 2020 Mar 12. PMID: 32166491.

-"Among these advancements is the AI-RAD companion, CNN, due to its high sensitivity" Honestly I can get the meaning: CNN is an AI-RAD companion??? Maybe the term CNN is not clear to the authors

-“(=1) and low false-positive nodule rate using 151 CT scan data, which has garnered significant attention" what do you mean by significant attention??

We have changed the above two comments with this statement: The AI-RAD companion, a CNN prototype, was able to automatically detect pulmonary nodules on LDCT with improved accuracy due to its high sensitivity (=1) and specificity(=0.708). Furthermore, these outcomes contributed to an enhanced ability to predict lung cancer(AUC-0.942)

"Additionally, the detection of 154 pulmonary nodules using AI was highest among standard-dose CT(AUC-0.989) com- 155 pared to low-dose (AUC-0.983) and very low-dose CT scans (AUC-0.970)" when talking about lung cancer screening, only LDCT is used, not standard CT. There are a lot of studies about the performance of DL-based CAD in lung nodule detection, why did the authors choose to talk about non-pertinent studies?

Thank you for the comment.  We appreciate the reviewers point; this paper aims to assess and compare the present data, where AI models are being studied against the standard of care to assess for any difference in outcomes which is why some “non-pertinent” data may be present.

"DL-based detection algorithm (DLAD) " do you mean DL based CAD?

 Yes, we have rectified it.

--" conventional methods among high-risk populations." I cannot get if conventional methods involve machine learning or radiological assessment

--"nodule patches"?

--"utilizing 2CT features: the bounding box anterior-posterior dimension (BB-AP) and the standard deviation of inverse difference moment (SD-IDM)." These are not CT features

Thank you for the correction: We have changed all of above three comments by removing the statements and explaining it with this statement below: Multi-scale CNN models relying solely on raw nodule patches, without proper definition of the morphology, were developed to capture nodule heterogeneity and it has achieved 88.84% accuracy for nodule classification against noisy background.

"Lung nodule segmentation plays a crucial role to clearly distinguish the areas and limits of the lung field from surrounding thoracic tissue" this sentence has no meaning

 Thank you for your comments, we have edited that with: Nodule segmentation is an image analysis technique that involves distinguishing and outlining the boundaries of lung nodules from the surrounding thoracic tissue.

"Manual segmentation of the airway is tedious" if you are working on lung nodules, you don't have to segment airways but just the nodules

 Thank you for the correction, we have made the changes.

"Although CNN is considered the gold standard for nodule segmentation" this is not true, support with references

Thank you for the comment, it was a language error leading to the misconception which we have rectified.

Non-Imaging Techniques:

When talking about lung K screening, non-imaging models should include not only blood tests but also the history and amount of smoke (current or not), emphysema, and family history... these factors are essential and not mentioned in the manuscript

Thank you for such a great comment, while this is an important point to be addressed and added, as per our literature review, there are no existing studies that incorporate the patient’s history with AI models to add to the current review. Given your comment, we have added that in our future perspective section.

"Factors like tumor location, pathology type, metastasis presence, and complications make diagnosis challenging" what do you mean

Overall staging of lung cancer is multi-factorial and training the AI model to incorporate that will need further training is what the authors are hoping to convey. We have cited one study supporting this ideology: Additionally, the incorporation of clinical parameters (such as symptoms, risk factors, smoking, and kitchen environment) resulted in an increased prediction rate of 87.3% [41]

"AI in lung cancer detection" Are you still talking about screening or not? 

We have changed it to diagnosis for simplicity.

"It involves using chest images," chest images are a diagnostic technique.

" MRI," MRI is not the gold standard for lung K investigation

Thank you for your comments, we have removed both the imaging modalities as they are not standard modes of diagnosis and staging.

", low dose computed tomography (LDCT)" use the abbreviation for the whole text

 We have removed it, thank you for the comment.

"The prompt staging of lung cancer requires the use of imaging techniques which 314 include computed tomography (CT) and positron emission tomography (PET) and CXR." Nobody use CXR for lung k staging

Thank you for your comments, we have removed the above statement.

" Lung nodules are the most prevalent imaging signs of lung cancer in the early stages, which makes manual CT scan reading more challenging" repetitive and already stated previously

Thank you for your comments, we have removed the above statement.

"AI might accelerate the accurate staging of lung cancer and curb the time-consuming tasks of reading pathology slides, CT scans, and CXR. The use of AI as a second reader for PET, CT, and CXR" again, CXR is not a technique for lung k staging

We understand, while we are aware it is not a method of lung staging, there are studies utilizing AI models with CXR which is why we initially mentioned them, however, we understand as a reader it can be confusing, thus we have removed them.

"Using Convolutional neural networks (CNN) as a model," again, you have already used in the text CNN different times

We have rectified that, thank you for the corrections.

Reviewer 2 Report

Comments and Suggestions for Authors

1. It has come to attention that a particular article has been cited multiple times, each time assigned a distinct reference number.

2. The article exhibits a loose structure, with unclear segmentation in its subtitles and sections.

3. I highly recommend addressing the variations in detecting solid and ground glass pulmonary nodules, examining their abilities, yield rates, and area under the curve (AUC) in diagnostic performance.

4. I recommend that the author consider including a section dedicated to discussing the limitations of current AI models or potential future developments. This section would provide an opportunity for the authors to share their perspectives with the readers regarding this topic.

Author Response

1. It has come to attention that a particular article has been cited multiple times, each time assigned a distinct reference number.

 We have found the article that was cited twice, we have removed that. Thank you for the comments.

2. The article exhibits a loose structure, with unclear segmentation in its subtitles and sections.

 We understand that the lack of change in font may have deterred the reader from headings and sub-headings, for that reason we have made edits with different font size to help the reader.

We have also included a figure to help the reader understand the format of our paper.

3. I highly recommend addressing the variations in detecting solid and ground glass pulmonary nodules, examining their abilities, yield rates, and area under the curve (AUC) in diagnostic performance.

We thank you for pointing that out, we have added the % values, AUC as appropriate to the paper. If some of our currently cited papers were missing that information, we have added some additional papers to support our review.

4. I recommend that the author consider including a section dedicated to discussing the limitations of current AI models or potential future developments. This section would provide an opportunity for the authors to share their perspectives with the readers regarding this topic.

Thank you for this great comment, we have added a limitation section to help address what is missing and support future developments.

Reviewer 3 Report

Comments and Suggestions for Authors

This study aims to review cases of various artificial intelligence methods related to lung cancer and find insights. In order to do this effectively, the following method of systematic literature review should be followed. The current level is simply enumerating past studies based on the classification criteria set by the researcher.

1. Please add a PRISMA flow diagram. In other words, indicate which keywords were used to search for related literature, and if there is excluded literature, write the criteria. And please use the diagram to present it.

2. Classify the contents of previous studies according to various dimensions such as diagnosis, treatment, data type, artificial intelligence technique, etc., and present them in tables or figures.

3. For readers who are not artificial intelligence experts, we expect a kind explanation of the various artificial intelligence techniques mentioned in this study.

Comments on the Quality of English Language

Minor corrections are needed, such as articles (a/the) and prepositions

Author Response

This study aims to review cases of various artificial intelligence methods related to lung cancer and find insights. In order to do this effectively, the following method of systematic literature review should be followed. The current level is simply enumerating past studies based on the classification criteria set by the researcher.

  1. Please add a PRISMA flow diagram. In other words, indicate which keywords were used to search for related literature, and if there is excluded literature, write the criteria. And please use the diagram to present it.

Thank you for your comments, we have added the methods section to the article. We would like to bring to the attention of the reviewer that the article is rather a comprehensive systematic review without any formal analysis. Hence, we have mentioned our search strategy for our readers to better understand the use of our article in their practice.

  1. Classify the contents of previous studies according to various dimensions such as diagnosis, treatment, data type, artificial intelligence technique, etc., and present them in tables or figures.

Thank you for the comment, we have added another table for diagnosis in addition to the treatment modalities of AI use in Lung Cancer. We have also added a figure explaining the subheadings of the article.

  1. For readers who are not artificial intelligence experts, we expect a kind explanation of the various artificial intelligence techniques mentioned in this study.

We understand that the goal of the article is to help the regular clinician understand the advances made in AI and for the more expert reader to stay up to date with the current available data. In our introduction section, we have briefly explained the major types of AI models, however, the basics of AI and its sub-types and its explanation remains beyond the scope of this article.

Reviewer 4 Report

Comments and Suggestions for Authors

This paper reviews the utility of artificial intelligence (AI) on lung cancer outcomes. While fairly comprehensive, some issues might be considered:

1. There exist a fair number of related prior narrative/comprehensive reviews largely concerned about lung cancer and AI, e.g.:

Rabbani, M., Kanevsky, J., Kafi, K., Chandelier, F., & Giles, F. J. (2018). Role of artificial intelligence in the care of patients with nonsmall cell lung cancer. European journal of clinical investigation, 48(4), e12901.

Sakamoto, T., Furukawa, T., Lami, K., Pham, H. H. N., Uegami, W., Kuroda, K., ... & Fukuoka, J. (2020). A narrative review of digital pathology and artificial intelligence: focusing on lung cancer. Translational Lung Cancer Research, 9(5), 2255.

Li, J., Wu, J., Zhao, Z., Zhang, Q., Shao, J., Wang, C., ... & Li, W. (2021). Artificial intelligence-assisted decision making for prognosis and drug efficacy prediction in lung cancer patients: A narrative review. Journal of Thoracic Disease, 13(12), 7021.

de Margerie-Mellon, C., & Chassagnon, G. (2022). Artificial intelligence: A critical review of applications for lung nodule and lung cancer. Diagnostic and Interventional Imaging.

etc.

It might be strongly considered to survey existing related reviews, and distinguish the current manuscript from those prior work.

2. The selection methodology of papers to be included in the comprehensive review might be briefly discussed, if possible. Was a systematic search of the literature performed, or were the papers informally curated/obtained by expert opinion, etc.?

3. A single table (Table 1) is presented, to compare studies on implementation of AI for treatment of lung cancers. It might be considered to include similar tables for the other sections, where appropriate.

Author Response

This paper reviews the utility of artificial intelligence (AI) on lung cancer outcomes. While fairly comprehensive, some issues might be considered:

  1. There exist a fair number of related prior narrative/comprehensive reviews largely concerned about lung cancer and AI, e.g.:

Rabbani, M., Kanevsky, J., Kafi, K., Chandelier, F., & Giles, F. J. (2018). Role of artificial intelligence in the care of patients with nonsmall cell lung cancer. European journal of clinical investigation, 48(4), e12901.

Sakamoto, T., Furukawa, T., Lami, K., Pham, H. H. N., Uegami, W., Kuroda, K., ... & Fukuoka, J. (2020). A narrative review of digital pathology and artificial intelligence: focusing on lung cancer. Translational Lung Cancer Research, 9(5), 2255.

Li, J., Wu, J., Zhao, Z., Zhang, Q., Shao, J., Wang, C., ... & Li, W. (2021). Artificial intelligence-assisted decision making for prognosis and drug efficacy prediction in lung cancer patients: A narrative review. Journal of Thoracic Disease, 13(12), 7021.

de Margerie-Mellon, C., & Chassagnon, G. (2022). Artificial intelligence: A critical review of applications for lung nodule and lung cancer. Diagnostic and Interventional Imaging.

etc.

It might be strongly considered to survey existing related reviews and distinguish the current manuscript from those prior work.

We understand that with any comprehensive review, there is a chance that a similar paper might have been published. To support the reasoning of our authors working on this project, firstly, we have summarized the role of AI in screening, diagnosis, and treatment in lung cancer overall, and also, we have updated our study with the newer literature to help clinicians make assessment and understand the advances already made in the field of AI in lung cancer. We urge the esteemed reviewer to please consider our project accordingly.

  1. The selection methodology of papers to be included in the comprehensive review might be briefly discussed, if possible. Was a systematic search of the literature performed, or were the papers informally curated/obtained by expert opinion, etc.?

Thank you for your comments, we have added the methods section to the article. We would like to bring to the attention of the reviewer that the article is rather a comprehensive systematic review without any formal analysis. Hence, we have mentioned our search strategy for our readers to better understand the use of our article in their practice.  

  1. A single table (Table 1) is presented, to compare studies on implementation of AI for treatment of lung cancers. It might be considered to include similar tables for the other sections, where appropriate.

We appreciate your comments and reasoning to add more figures, we have added another table to summarize diagnostic methods and another figure to understand the flow of the paper and which areas has implementation of AI.

Round 2

Reviewer 1 Report

Comments and Suggestions for Authors

Machine learning is the remarkable line 69. You have just used the abbreviation ML, but you repeat the extended-expression

"We explored the database using the following Medical Subject Heading 93 (MeSH) terms: Artificial Intelligence, Machine learning, Radiomics, Deep learning,... We included all the prospective, retrospective, and review papers, with the main emphasis on the use and implementation of artificial intelligence on lung cancer."

If the author performed the research using these keywords separately, the references included in the manuscript should have a significantly higher number.

", each with an independent 97 perspective" this is too vague, what do you mean?

there is no mention about the selection of the articles included in the review

"techniques under two subheadings:: Imaging (nodule detection, segmentation, and characterization) and Non-imaging techniques" the decision to include so different topics should be explained

"These nodules are abnormal growths in the lung, which are potential can serve as early indicators of lung cancer and can be detected and characterized using CXR and CT scans"

"Nodule segmentation plays a crucial role to clearly distinguish is an image analysis technique that distinguishes and limits ... from the surrounding thoracic tissue" This is only a partial explanation, as nodule segmentation should also assess at least the volume of the lesion (and different methods also the densitometric characteristics)

"it's" it is

"AI in Predicting Treatment Outcomes" in this section the ref numbers are strange (8280,81,82)

"One of the limitations of AI is the lack of large datasets of clinical data to train the model,  which can hinder its performance." This is not true. Many public datasets are available for lung nodules detection and characterization (e.g., LIDC-IDRI, LUNA 16, Ali Tianchi)

Comments on the Quality of English Language

Moderate

Reviewer 4 Report

Comments and Suggestions for Authors

We thank the authors for largely addressing our previous comments. Some references (e.g. 70, 7169,70 in Line 372) might be rechecked.

Comments on the Quality of English Language

N/A